# Skullcapflavone II Inhibits Degradation of Type I Collagen by Suppressing MMP-1 Transcription in Human Skin Fibroblasts

**DOI:** 10.3390/ijms20112734

**Published:** 2019-06-04

**Authors:** Young Hun Lee, Eun Kyoung Seo, Seung-Taek Lee

**Affiliations:** 1Department of Biochemistry, College of Life Science and Biotechnology, Yonsei University, Seoul 03722, Korea; isop_tera@naver.com; 2Graduate school of Pharmaceutical Sciences, College of Pharmacy, Ewha Womans University, Seoul 03790, Korea; yuny@ewha.ac.kr

**Keywords:** extracellular matrix, fibroblast, inflammation, MMP-1, skullcapflavone II, type I collagen

## Abstract

Skullcapflavone II is a flavonoid derived from the root of *Scutellaria baicalensis*, a herbal medicine used for anti-inflammatory and anti-cancer therapies. We analyzed the effect of skullcapflavone II on the expression of matrix metalloproteinase-1 (MMP-1) and integrity of type I collagen in foreskin fibroblasts. Skullcapflavone II did not affect the secretion of type I collagen but reduced the secretion of MMP-1 in a dose- and time-dependent manner. Real-time reverse transcription-PCR and reporter gene assays showed that skullcapflavone II reduced MMP-1 expression at the transcriptional level. Skullcapflavone II inhibited the serum-induced activation of the extracellular signal-regulated kinase (ERK) and c-Jun N-terminal kinase (JNK) signaling pathways required for MMP-1 transactivation. Skullcapflavone II also reduced tumor necrosis factor (TNF)-α-induced nuclear factor kappa light chain enhancer of activated B cells (NF-κB) activation and subsequent MMP-1 expression. In three-dimensional culture of fibroblasts, skullcapflavone II down-regulated TNF-α-induced MMP-1 secretion and reduced breakdown of type I collagen. These results indicate that skullcapflavone II is a novel biomolecule that down-regulates MMP-1 expression in foreskin fibroblasts and therefore could be useful in therapies for maintaining the integrity of extracellular matrix.

## 1. Introduction

Skullcapflavone II, a naturally occurring flavonoid compound with a polyphenolic structure also known as neobaicalein, is derived from the roots of *Scutellaria baicalensis*, *S. litwinowii*, and *S. pinnatifida* [1,2,3]. Biological functions attributed to skullcapflavone II include reducing inflammation, inhibiting osteoclastogenesis, decreasing cell growth, inducing apoptosis, and down-regulating cholesterol [2,4,5,6,7]. Skullcapflavone II inhibits ovalbumin-induced airway inflammation via a decrease in transforming growth factor-β1 expression and subsequent decrease in SMAD2/3 activation [4]. Skullcapflavone II inhibits osteoclastogenesis with reduced activation of mitogen-activated protein kinases (MAPKs), Src, and cyclic adenosine monophosphate (cAMP) response element binding protein, and attenuates the survival and bone resorption function of osteoclasts via down-regulation of integrin signaling [5]. Skullcapflavone II inhibits cell proliferation in a variety of cancer cell lines, including LNCaP, PC-3, and HeLa [2,6]. In addition, skullcapflavone II reportedly inhibits the mRNA expression of proprotein convertase subtilisin/kexin type 9, which prevents the recycling of endocytosed low-density-lipoprotein receptors (LDLRs) to the cell surface, thereby increasing cell-surface LDLR levels and lowering plasma cholesterol levels [7]. 

Collagen is the primary structural protein in extracellular spaces in mammals and functions to strengthen and support various connective tissues, such as tendons, ligaments, bones, and skin [8,9]. The collagen family contain at least one triple-helical domain consisting of three α-chains with a repeating amino acid sequence (Gly-X-Y)_n_ [10]. A total of 28 types of collagen identified in humans can be divided into subfamilies based on their supramolecular assemblies; fibrils, beaded filaments, anchoring fibrils, and networks [11,12].

Collagen is degraded during various normal physiological processes involving tissue remodeling, such as organ morphogenesis, wound healing, and skin aging. In addition, collagen is degraded during numerous pathological conditions, such as inflammation, arthritis, atherosclerotic cardiovascular disease, and tumorigenesis [10,11]. Matrix metalloproteinases (MMPs) are zinc-dependent endopeptidases that play a major role in tissue remodeling processes by cleaving extracellular matrix components, including collagen [13,14]. MMP-1 cleaves the triple helix of fibril-forming collagens, including types I, II, and III; in type I collagen, it cleaves at Gly^775^↓Ile^776^ of the α1 chain and Gly^775^↓Leu^776^ of the α2 chain to generate 3/4- and 1/4-length fragments [15]. 

Collagen is also frequently degraded in inflammation lesions. In inflamed acne lesions, for example, collagen degradation is increased as a result of up-regulation of inflammatory cytokine and MMP expression [16]. Among the various collagen types, type I accounts for over 90% of all collagens in the human body [9] and is highly expressed in fibroblasts [17]. In addition, MMP-1 is expressed in unstimulated fibroblasts and is upregulated by inflammation [18,19]. Because skullcapflavone II exhibits anti-inflammatory activity, we investigated the effect of skullcapflavone II on the expression of MMP-1 and the integrity of type I collagen in fibroblasts. Specifically, we examined whether skullcapflavone II affects the production of type I collagen or MMP-1-mediated degradation of type I collagen. In addition, we analyzed the signaling pathways involved in skullcapflavone II-mediated suppression of MMP-1 expression. Finally, using three-dimensional (3D) culture of fibroblasts, we examined the effect of skullcapflavone II on breakdown of type I collagen.

## 2. Results

### 2.1. Skullcapflavone II Decreased MMP-1 Expression in Foreskin Fibroblasts

We investigated the effect of skullcapflavone II (Appendix A) on the secretion of MMP-1 and type I collagen in primary human foreskin fibroblasts and primary human buttock dermal fibroblasts. It was reported that human fibroblasts secrete MMP-1 as a pro-form; mostly unglycosylated (52 kDa) and partly glycosylated (57 kDa) [20]. We detected a major band of MMP-1 at 52 kDa in foreskin fibroblasts and buttock dermal fibroblasts (Figure 1A), suggesting that it should be proMMP-1. Skullcapflavone II significantly decreased the secretion of MMP-1 in a dose-dependent manner in both cell types but did not significantly affect the secretion of type I collagen (Figure 1A). Compared with untreated cells, foreskin fibroblasts secreted significantly lower amounts of MMP-1 in the presence of 3 μM skullcapflavone II, with the effect decreasing by 24 h in serum-free Dulbecco’s Modified Eagle’s Medium (DMEM) and by 48 h in DMEM supplemented with 3% fetal bovine serum (FBS) (Figure 1B).

To examine whether the reduction in MMP-1 secretion by cells treated with skullcapflavone II was associated with a decrease in cell proliferation, we monitored the growth of skullcapflavone II-treated foreskin fibroblasts using the 3-(4,5-dimethyl thiazol-2-yl)-2,5-diphenyltetrazolium bromide (MTT) assay. Cell growth was unaffected by skullcapflavone II at concentrations up to 3 μM but slightly decreased at a concentration of 10 μM (up to 15% decrease in growth at 2 days) (Figure 2A). Flow cytometry analyses indicated that the decrease in the growth of foreskin fibroblasts in the presence of 10 μM skullcapflavone II was not due to cytotoxic effects (Figure 2B). For subsequent experiments, therefore, we used 3 μM skullcapflavone II, a concentration that had no effect on cell growth.

### 2.2. Skullcapflavone II Decreased Transcription of the MMP-1 Gene in Foreskin Fibroblasts

To determine whether the observed skullcapflavone II-mediated inhibition of MMP-1 secretion was due to suppression of *MMP-1* transcription, we performed real-time RT-PCR and reporter gene assays using foreskin fibroblasts. Consistent with MMP-1 protein levels, *MMP-1* mRNA levels declined in a dose-dependent manner in cells treated with skullcapflavone II (Figure 3A). From the reference curve generated by serial dilution of *MMP-1* cDNA, the *MMP-1* mRNA molecules per cell were estimated to be 40.0 at 0 uM skullcapflavone II and 27.2 at 10 uM skullcapflavone II. Treatment with skullcapflavone II also reduced luciferase activity driven by the *MMP-1* promoter to 70% of the control (Figure 3B). These data demonstrate that skullcapflavone II inhibits expression of the *MMP-1* gene at the transcription level.

### 2.3. Skullcapflavone II Inhibited MMP-1 Expression by Blocking the Activation of Activator Protein-1 (AP-1)

*MMP-1* gene expression is positively regulated by the activation of extracellular signal-regulated kinase (ERK), c-Jun N-terminal kinase (JNK), p38 MAPK, and nuclear factor kappa light chain enhancer of activated B cells (NF-κB) signaling pathways [21,22]. To investigate how skullcapflavone II suppresses MMP-1 transactivation, phosphorylation of various signaling proteins was analyzed in foreskin fibroblasts incubated with and without skullcapflavone II. Serum-induced tyrosine phosphorylation of cellular proteins was significantly decreased in a dose-dependent manner in cells treated with skullcapflavone II (Figure 4A). The presence of FBS strongly enhanced the phosphorylation of ERK1/2, moderately enhanced the phosphorylation of JNK, and weakly enhanced the phosphorylation of p38 MAPK (Figure 4B). Treatment with skullcapflavone II reduced the serum-induced phosphorylation of ERK1/2 and JNK in a dose-dependent manner but had no effect on the phosphorylation of p38 MAPK (Figure 4B). Phosphorylation of NF-κB p65 was not affected by the presence of FBS and did not decrease in the presence of skullcapflavone II. In addition, skullcapflavone II significantly decreased the serum-enhanced phosphorylation of c-Jun (Figure 4C). These findings demonstrate that skullcapflavone II down-regulates *MMP-1* expression by reducing activation of the ERK and JNK pathways and thus activation of the transcription factor AP-1, which plays an important role in MMP-1 transactivation.

### 2.4. Skullcapflavone II Inhibited Tumor Necrosis Factor (TNF)-α-Induced MMP-1 Expression by Blocking Activation of NF-κB

The pro-inflammatory cytokine TNF-α up-regulates MMP-1 expression in diverse cell types, including dermal fibroblasts [23,24,25]. As expected, stimulation of foreskin fibroblasts with TNF-α significantly up-regulated MMP-1 secretion (Figure 5A). Treatment with skullcapflavone II decreased MMP-1 expression induced by TNF-α. However, consistent with the effect of serum stimulation, skullcapflavone II treatment did not affect the secretion of type I collagen in TNF-α-treated cells (Figure 5A).

Since TNF-α is known to activate NF-κB signaling [26], we investigated whether skullcapflavone II suppresses NF-κB activation for the down-regulation of MMP-1. Treatment with skullcapflavone II also reduced the TNF-α-induced phosphorylation of NF-κB p65 (Figure. 5B). Although stimulation of cells with TNF-α in serum-free medium for 24 h did not lead to a significant increase in phosphorylation of ERK1/2 and JNK, skullcapflavone II reduced the phosphorylation of ERK1/2 with or without TNF-α stimulation (Figure 5B). Moreover, phosphorylation of ERK1/2 and JNK was increased by stimulation with TNF-α for 10 min in serum-free medium but significantly decreased by treatment with skullcapflavone II (Appendix A). These findings suggest that skullcapflavone II down-regulates TNF-α-induced MMP-1 expression by reducing the activation of NF-κB and AP-1.

### 2.5. Skullcapflavone II Decreased the Breakdown of Type I Collagen in 3D Culture of Foreskin Fibroblasts

MMP-1 is a central enzyme involved in the degradation of type I collagen. Since proMMP-1 secreted from fibroblasts in 2D culture is not activated into mature active form (Figure 1A), it is difficult to detect cleavage of type I collagen by MMP-1 in the medium. We therefore examined whether skullcapflavone II affects the degradation of type I collagen by down-regulating MMP-1 expression using 3D culture of foreskin fibroblasts. Consistent with two-dimensional culture conditions, skullcapflavone II down-regulated the TNF-α-induced secretion of MMP-1 in 3D culture (Figure 6A). Experiments using collagen type I cleavage-site antibody revealed that TNF-α stimulation increased the generation of cleaved 3/4 fragments of type I collagen. Interestingly, treatment with skullcapflavone II significantly decreased the amount of cleaved 3/4 fragments of type I collagen in 3D culture of foreskin fibroblasts (Figure 6B). These data demonstrate that skullcapflavone II inhibits collagenolysis via down-regulation of MMP-1 expression.

## 3. Discussion

Roots of *Scutellaria baicalensis*, *S. litwinowii*, and *S. pinnatifida* are used as herbal medicines due to their anti-tumorigenic, anti-fibrotic, anti-inflammatory, and antioxidant effects [2,3,4,5,6,27,28]. Skullcapflavone II is one of the common components of *Scutellaria* roots, and it is considered a potential therapeutic compound for use in treating cancer, allergic asthma, and bone diseases. In this study, we examined whether skullcapflavone II regulates the expression of type I collagen and the enzyme MMP-1, which degrades type I collagen in fibroblasts. 

We found that skullcapflavone II decreased the secretion of MMP-1 but had no effect on the secretion of type I collagen by foreskin fibroblasts and buttock dermal fibroblasts. Skullcapflavone II did not affect the proliferation of foreskin fibroblasts at concentrations up to 3 μM but inhibited proliferation at a concentration of 10 μM. However, at a concentration of 10 μM, skullcapflavone II did not induce apoptosis of foreskin fibroblasts. Our findings thus suggest that, at higher concentrations, skullcapflavone II reduces not only the secretion of MMP-1 by fibroblasts but also their proliferation.

We found that skullcapflavone II inhibits the expression of MMP-1 at the transcriptional level. Induction of MMP-1 transcription depends on activation of the transcription factors AP-1 and NF-κB. The *MMP-1* promoter has three AP-1 binding sites, at −70, −186, and −1602 bp [22,29], and an NF-κB binding site at −2886 bp [30,31], upstream of the transcriptional start site. AP-1 is composed of the polypeptides c-Jun and c-Fos. Activation of JNK subsequently leads to c-Jun phosphorylation, whereas activation of ERK induces c-Fos transactivation [32,33]. In our study, serum (i.e., FBS) stimulation induced tyrosine phosphorylation of cellular proteins and phosphorylation of ERK, JNK, p38, and c-Jun. Skullcapflavone II treatment decreased the serum-induced phosphorylation of cellular proteins, ERK, JNK, and c-Jun. These results demonstrate that skullcapflavone II reduces MMP-1 expression by suppressing AP-1 activation under serum-induced conditions. 

In addition to inducing MMP-1 expression, AP-1 also up-regulates cell proliferation [32,33]. It is interesting to note that, in our experiments, skullcapflavone II reduced cell proliferation at higher concentrations but did not induce apoptosis. As we demonstrated here that skullcapflavone II suppresses the activation of AP-1, higher concentrations of skullcapflavone II could slow progression of the cell cycle without causing cell death.

MMP-1 expression is known to be up-regulated by a variety of inflammatory cytokines and growth factors, such as TNF-α, interleukins-1, -4, -5, -6, -8, and -10, fibroblast growth factors-1, -2, -7, and -9, epidermal growth factor (EGF), and platelet-derived growth factor [34,35]. Dermal fibroblasts are often subjected to inflammatory conditions associated with infections involving bacteria, viruses, or fungi or as a result of ultraviolet or ionized radiation exposure. As expected, treatment of fibroblasts with TNF-α to mimic an inflammatory state resulted in up-regulated secretion of MMP-1 and increased phosphorylation of NF-κB p65 as well as ERK1/2 and JNK. ERK and JNK were activated following short-term stimulation with TNF-α, and activation of NF-κB p65 was sustained with long-term TNF-α stimulation. Skullcapflavone II decreased TNF-α-induced MMP-1 secretion and also decreased phosphorylation of NF-κB p65, ERK1/2, and JNK. These results suggest that skullcapflavone II inhibits MMP-1 transactivation by suppressing both the AP-1 and NF-κB signaling pathways under pro-inflammatory conditions. 

Skullcapflavone II also reportedly has antioxidant activity [5]. Reactive oxygen species (ROS) such as H_2_O_2_ and singlet oxygen (^1^O_2_) can activate NF-κB under a variety of circumstances [36,37,38]. For example, ROS activate IκB kinase by phosphorylation of its subunits on serine and tyrosine residues of the activation loops [39]. ROS also activate NF-κB through tyrosine phosphorylation of IκBα, without its degradation [40]. In addition, ROS can activate AP-1 under a variety of physiological conditions such as inflammation and tumorigenesis [41,42]. ROS activate protein tyrosine kinases by specific oxidation of cysteine SH groups, inducing an activating conformational change in the enzymes [43,44]. Thus, ROS induce autophosphorylation of receptor protein tyrosine kinases such as the EGF receptor [43] and tyrosine phosphorylation of downstream signaling proteins such as Src [45], thus enhancing activation of AP-1. We demonstrated that skullcapflavone II decreases cellular ROS generation (data not shown) and suppresses the TNF-α-induced activation of NF-κB and serum-induced tyrosine phosphorylation of cellular proteins and activation of ERK1/2 and JNK. Therefore, we hypothesized that the decreased activation of NF-κB and AP-1 by skullcapflavone II is mediated, at least in part, by its anti-oxidant activity.

Breakdown of collagen by MMP-1 is a critical process in the regulation of tissue remodeling, development, and morphogenesis [12,46]. Fibroblasts grown in 2D culture secrete the pro-form of MMP-1 which cannot cleave type I collagen. To characterize the effect of skullcapflavone II on the breakdown of type I collagen, we used an in vivo-mimicking 3D culture system of foreskin fibroblasts with an anti-type I collagen cleavage site antibody that enables detection of cleaved 3/4 fragments of type I collagen [47,48]. We found that skullcapflavone II decreased TNF-α-induced MMP-1 expression and degradation of type I collagen in 3D culture of foreskin fibroblasts. Therefore, we assume that proMMP-1 is at least partially processed into mature MMP-1 in a 3D culture system. Based on these results, we believe that skullcapflavone II could be useful in therapies aimed at maintaining the integrity of the extracellular matrix or for treating aging-induced and inflammation-related deterioration of the extracellular matrix.

Some natural flavonoids, such as quercetin, kaempferol, wogonin, apigenin, and luteolin, have also demonstrated efficacy at suppressing MMP-1 expression by reducing AP-1 activation [49,50]. However, relatively high concentrations (>10 μM) of these flavonoids are required to suppress MMP-1 expression in human dermal fibroblasts [49]. In our study, much lower concentrations (≤3 μM) of skullcapflavone II were sufficient to significantly down-regulate the expression of MMP-1. In addition, skullcapflavone II has methoxy (O–CH_3_) groups at positions 6, 7, and 8 of the A ring and the 6’-position of the B ring, which is functionally important because it has been reported that polymethoxyflavones pass through the cell membrane and are readily transported via the blood circulation [51,52]. Therefore, we hypothesize that these O–CH_3_ groups play an important role in the greater bioavailability of skullcapflavone II compared to other flavonoids.

In summary, this is the first study demonstrating the inhibitory effect of skullcapflavone II on the expression of MMP-1 and degradation of type I collagen in foreskin fibroblasts (Figure 7). We propose that skullcapflavone II would be a useful chemopreventive compound for treating physiological conditions associated with up-regulation of MMP-1 and the loss of extracellular matrix integrity, such as skin aging.

## 4. Materials and Methods

### 4.1. Reagents and Antibodies

Skullcapflavone II (5-hydroxy-2-[2-hydroxy-6-methoxyphenyl]-6,7,8-trimethoxychromen-4-one) was purchased from ChemFaces (Wuhan, Hubei, China). Antibodies against phospho-ERK, ERK2, p38, and NF-κB p65 were purchased from Santa Cruz Biotechnology (Santa Cruz, CA, USA). Antibodies against phospho-NF-κB p65 (Ser536), phospho-JNK, JNK, phospho-p38, phospho-c-Jun, and c-Jun were purchased from Cell Signaling Technology (Danvers, MA, USA). Anti-phospho-tyrosine antibody (clone 4G10) was purchased from Millipore (Billerica, MA, USA). Anti-collagen type I cleavage-site antibody was purchased from ImmunoGlobe (Himmelstadt, Germany). Anti-GAPDH antibody was purchased from AbClone (Seoul, Korea). Horseradish peroxidase-conjugated goat anti-mouse IgG and rabbit IgG were obtained from KOMA Biotech (Seoul, Korea). Anti-MMP-1 and pro-collagen α1(I) N-propeptide (pN-ColIα1) antibodies were a gift from Dr. Chung, J. H. (Seoul National University College of Medicine, Republic of Korea) [53]. Alexa Fluor^®^ 488 goat anti-rabbit IgG (H+L) and rhodamine-conjugated phalloidin were purchased from Thermo Fisher Scientific (Waltham, MA, USA).

### 4.2. Cloning of the Human MMP-1 Promoter in a Reporter Plasmid

To generate a reporter construct of the human *MMP-1* promoter (GenBank Accession No. NM_000011.10), a 1938-bp DNA fragment including the promoter of the human *MMP-1* gene (−1880 to +40) was PCR-amplified using genomic DNA from human dermal fibroblasts as a template, PrimeSTAR^®^ GXL DNA polymerase (TaKaRa, Shiga-ken, Japan), and the primer pair 5’-GAA*GCTAGC*TCCCTCACAGTCGAGTATATCTGCCAC-3’, which includes a *Nhe*I site (italicized) and 5’-GAA*AAGCTT*GCAAGGTAAGTGATGGCTTCCCAG-3’, which includes a *Hin*dIII site (italicized). The PCR product cleaved with *Nhe*I and *Hin*dIII was cloned into the pGL3-Basic luciferase reporter (Promega, Madison, WI, USA) that was digested with *Nhe*I and *Hin*dIII to generate the pGL3-*MMP-1* promoter.

### 4.3. Cell Culture

A primary culture of human foreskin fibroblasts was obtained from Welskin (Seoul, Korea). A primary culture of human dermal fibroblasts derived from a buttock skin of a young individual was provided by Chung, J. H. (Seoul National University College of Medicine, Seoul, Korea). Cells were maintained sub-confluently in DMEM (Gibco/Thermo Fisher Scientific, Waltham, MA, USA) supplemented with 10% FBS (Gibco/Thermo Fisher Scientific), 100 U/mL penicillin, and 100 µg/mL streptomycin at 37 °C in an atmosphere of 5% CO_2_ and 95% air. Cells were plated on culture dishes and incubated overnight for attachments. The passage numbers for foreskin fibroblasts or buttock dermal fibroblasts were 13–18 or 8–13, respectively.

### 4.4. RNA Isolation and Reverse Transcription (RT)-PCR Analysis

Total RNA was isolated from foreskin fibroblasts using TRIzol reagent (Invitrogen, Carlsbad, CA, USA) as described previously [54]. cDNA was synthesized from total RNA (2 μg) using AMV RT system (Promega, Madison, WI, USA) and oligo (dT)_15_ primer. Real-time PCR was conducted using a QuantiTect SYBR Green PCR kit (Qiagen, Hilden, Germany) and the QuantStudio 3 Real-Time PCR system (Applied Biosystems, Foster City, CA, USA). Primer sequences and annealing temperatures are described in Appendix A.

### 4.5. Preparation of Conditioned Media and Cell Lysates, and Western Blot Analysis

Sub-confluent cells were incubated with serum-free medium for 24 h, and the resulting conditioned medium was collected by centrifugation at 2000× *g* for 3 min. Cell pellets were lysed with 1× SDS sample buffer for analysis of GAPDH or radio-immunoprecipitation assay (RIPA) lysis buffer (50 mM Tris-HCl, pH7.4, 150 mM NaCl, 1% NP-40, 0.5% sodium deoxycholate, and 0.1% SDS) containing 1 mM NaF, 1 mM NA_3_VO_4_, and a SIGMAFAST^TM^ protease inhibitor tablet (Sigma-Aldrich, St. Louis, MO, USA) for analysis of signaling proteins. Western blot analysis was performed as described previously [55,56]. The MultiGauge software (Fujifilm, Tokyo, Japan) was used to quantify the band intensities. 

### 4.6. Cell Growth Assay

Cell growth was analyzed as previously described [57]. Foreskin fibroblasts (0.5 × 10^4^ cells/well) were plated in 96-well plates and incubated in medium supplemented with 10% FBS and various concentrations of skullcapflavone II for up to 2 days. Viable cells were stained with MTT, solubilized with dimethyl sulfoxide (DMSO), and the absorbance was measured at 565 nm using a microplate reader (Molecular Devices, San Jose, CA, USA). 

### 4.7. Flow Cytometry

Foreskin fibroblasts (0.5 × 10^4^ cells/well) were seeded in 6-well plates and incubated in DMEM supplemented with 10% FBS and various concentrations of skullcapflavone II for 24 h. Briefly, cells were washed twice with cold PBS and then resuspended in binding buffer (10 mM HEPES, pH 7.4, 140 mM NaCl, and 2.5 mM CaCl_2_) to a final density of 1 × 10^6^ cells/mL. A total volume of 100 μL of detached cells, including 5 μL of FITC annexin V (BD Biosciences, Bedford, MA, USA) and 5 μL of 50 μg/mL 7-aminoactinomycin D (7-AAD, Invitrogen), was incubated for 15 min at room temperature. Apoptotic cells were then analyzed by flow cytometry (BD FACSCalibur, BD Biosciences, Bedford, MA, USA).

### 4.8. Dual-Luciferase Reporter Assay

Transfection of reporter genes into foreskin fibroblasts was conducted using Lipofectamine LTX (Thermo Fisher Scientific). Foreskin fibroblasts (5 × 10^4^ cells/well) were seeded in 24-well plates, and the medium was replaced with fresh DMEM supplemented with 10% FBS. Promoterless pGL3-Basic (0.36 μg) or pGL3-*MMP1* promoter (0.5 μg) encoding firefly luciferase driven by *MMP-1* promoter and pRL-TK (0.05 μg, Promega, Madison, WI, USA) encoding *Renilla* luciferase driven by herpes simplex virus thymidine kinase promoter in 25 μL Opti-MEM were incubated with Lipofectamine LTX (0.75 μL, Invitrogen, Carlsbad, CA, USA) and PLUS^TM^ reagent (0.25 μL, Invitrogen, Carlsbad, CA, USA) in 25 μL Opti-MEM (Gibco/Thermo Fisher Scientific, Waltham, MA, USA) for 20 min at room temperature. Cells were treated with this mixture for 5 h and then incubated with DMEM supplemented with 10% FBS for 24 h. The cells were then treated with skullcapflavone II or DMSO in serum-free medium for 12 h. Luciferase activity was measured using the dual-luciferase reporter assay system (Promega), and the firefly luciferase activity in transfected cells was normalized to the *Renilla* luciferase activity.

### 4.9. Collagenolysis in 3D Culture, Confocal Fluorescence Microscopy, and Image Acquisition

Foreskin fibroblasts (5 × 10^5^ cells/mL) were trypsinized and resuspended in 2.8 mg/mL of rat tail collagen I solution (Corning Inc., Corning, NY, USA):5× DMEM:10× reconstitution buffer (260 mM NaHCO_3_, 200 mM HEPES, and 50 mM NaOH) = 7:2:1, and then 0.15 mL of the cell mixture was placed in a glass-bottom (35 mm × 10 mm, hole 13 φ) dish (SPL Life Sciences, Gyeonggi-do, Korea). After solidifying for 1 h at 37°C, 2 mL of phenol red-free DMEM (Hyclone, South Logan, UT, USA) with or without TNF-α and/or skullcapflavone II was added, and collagen-embedded cells were incubated for 24 h at 37 °C in an atmosphere of 5% CO_2_ and 95% air. For nucleus staining, cells were incubated with Hoechst 33258 (2 μg/mL) for 30 min and then fixed in 3.7% paraformaldehyde for 30 min, permeabilized in 0.2% Triton-X 100 for 10 min, blocked in 3% bovine serum albumin for 30 min, and immunostained overnight at 4 °C with rabbit anti-collagen type I cleavage-site antibody (2.5 μg/mL). The cells were then washed with PBS and incubated with Alexa Fluor^®^ 488 goat anti-rabbit IgG (H+L) (Invitrogen, Carlsbad, CA, USA) and phalloidin-rhodamine (1 U/mL). Images were obtained on a confocal microscope (LSM700; Carl Zeiss, Feldbach, Switzerland) with 20× Plan-Apochromat objective lens and Zen software (Carl Zeiss, Oberkochen, Germany). The excitation wavelengths were 405 nm for Hoechst 33258, 488 nm for Alexa Fluor^®^ 488, and 555 nm for rhodamine. To avoid bias during image acquisition, all images were obtained from randomly selected fields and acquired using the same parameters including exposure time, laser power, and offset settings. The intensity of fluorescence was determined using Image J software (National Institutes of Health, Bethesda, MD, USA). 

### 4.10. Statistical Analyses

All data are expressed as the mean ± S.D. of at least three independent experiments. Statistical significance was analyzed using the paired two-tailed Student’s *t*-test, except for the fluorescence image analysis using the unpaired two-tailed Student’s *t*-test. A *p*-value <0.05 was considered indicative of statistical significance.

## 5. Conclusions

We were able to show that skullcapflavone II inhibits the expression of MMP-1 and the degradation of type I collagen in foreskin fibroblasts. Skullcapflavone II suppresses transcription of MMP-1 mRNA through reduced activation of AP-1 and NF-κB. Skullcapflavone II also reduces type I collagen degradation in the 3D fibroblast culture. We propose that skullcapflavone II would be a useful chemopreventive compound for the treatment of physiological conditions associated with the up-regulation of MMP-1 and the loss of extracellular matrix integrity, such as skin aging.

## Figures and Tables

**Figure 1 ijms-20-02734-f001:**
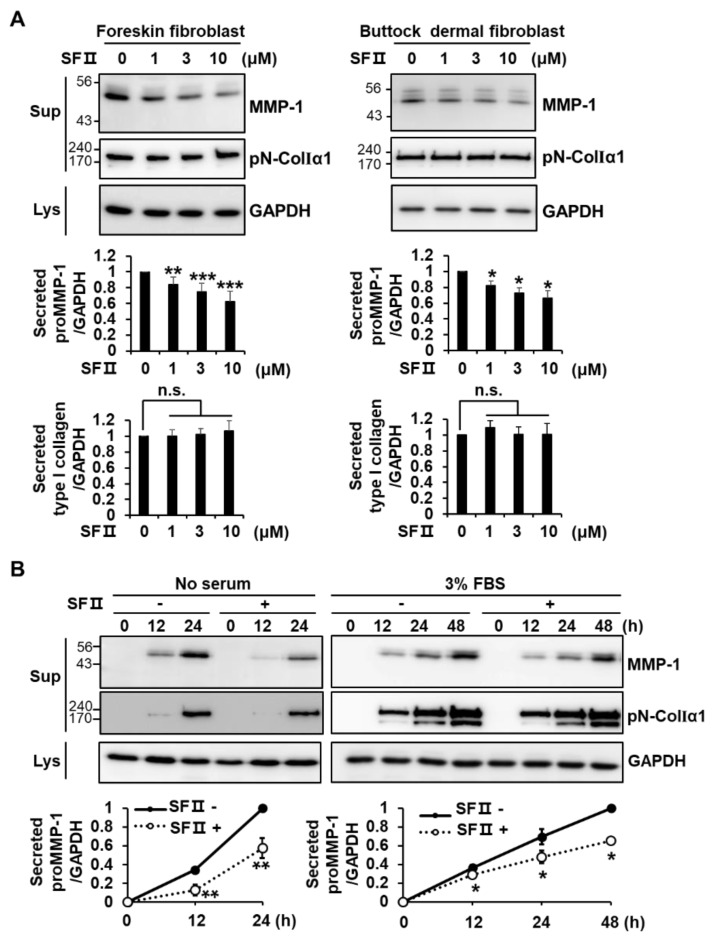
Effect of skullcapflavone II on expression of matrix metalloproteinase-1 (MMP-1) and type I collagen in fibroblasts. (**A**) Primary human foreskin fibroblasts and primary human buttock skin fibroblasts were incubated for 24 h in serum-free Dulbecco’s Modified Eagle’s Medium (DMEM) with the indicated concentrations of skullcapflavone II. * *p* < 0.05, ** *p* < 0.01, and *** *p* < 0.001 vs. the sample incubated with 0 μM skullcapflavone II; n.s., not significant. (**B**) Foreskin fibroblasts were incubated for the indicated times in serum-free DMEM or DMEM containing 3% fetal bovine serum (FBS) with (+) or without (−) 3 μM skullcapflavone II. Conditioned medium and cell lysates were analyzed by 9% SDS-PAGE and Western blot with anti-MMP-1, anti-pN-ColIα1, and anti-glyceraldehyde 3-phosphate dehydrogenase (GAPDH) antibodies. * *p* < 0.05 and ** *p* < 0.01 vs. the sample incubated without skullcapflavone II.

**Figure 2 ijms-20-02734-f002:**
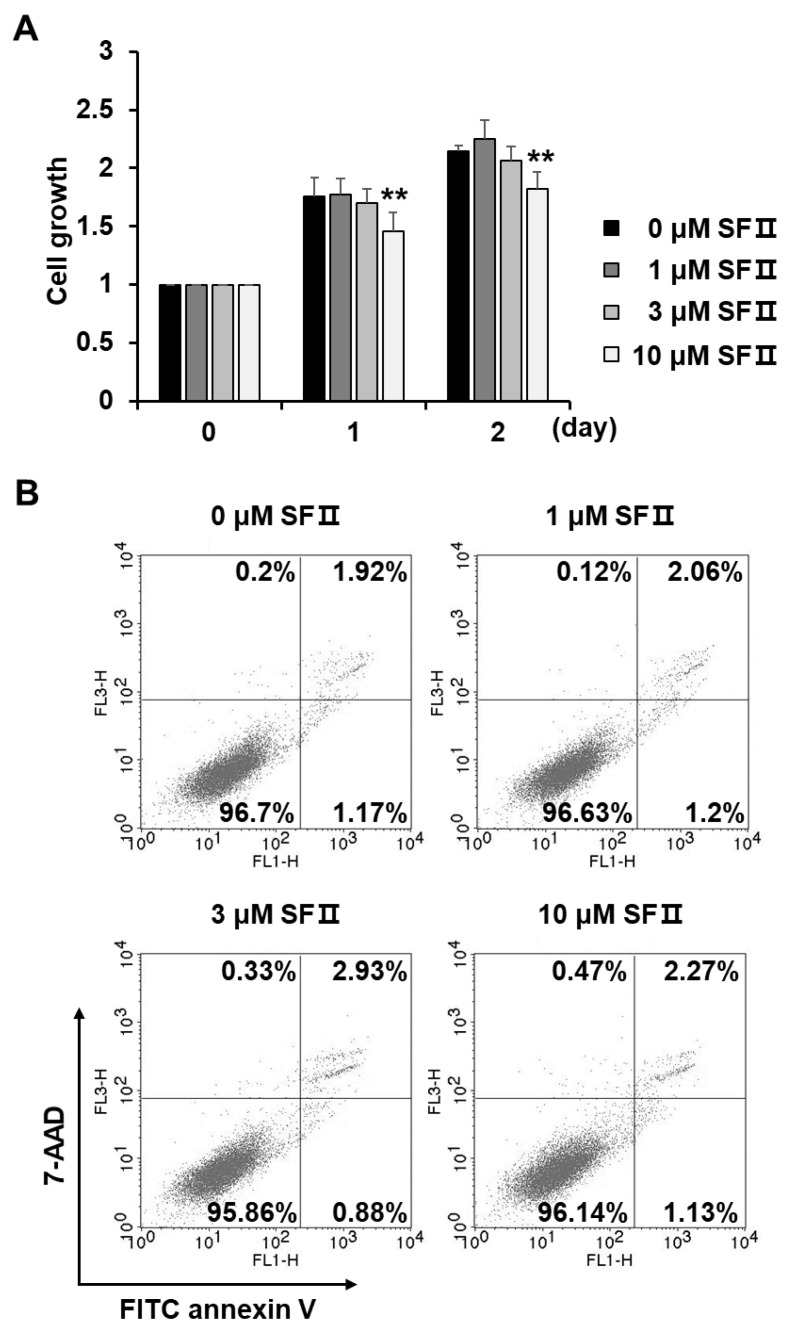
Effect of skullcapflavone II on proliferation and cytotoxicity of foreskin fibroblasts. Sub-confluent foreskin fibroblasts were plated in DMEM containing 10% FBS and incubated with indicated concentrations of skullcapflavone II. (**A**) The number of viable cells was measured based on the absorbance at a wavelength of 565 nm using 3-(4,5-dimethyl thiazol-2-yl)-2,5-diphenyltetrazolium bromide (MTT) reagent. The number of viable cells in the presence of skullcapflavone II is shown as fold change relative to that in the absence of skullcapflavone II. ** *p* < 0.01 vs. the sample incubated with 0 μM skullcapflavone II. (**B**) After 24 h of incubation with skullcapflavone II, apoptotic cells were stained with fluorescein isothiocyanate (FITC) annexin V and 7-aminoactinomycin (7-AAD) and then analyzed by flow cytometry.

**Figure 3 ijms-20-02734-f003:**
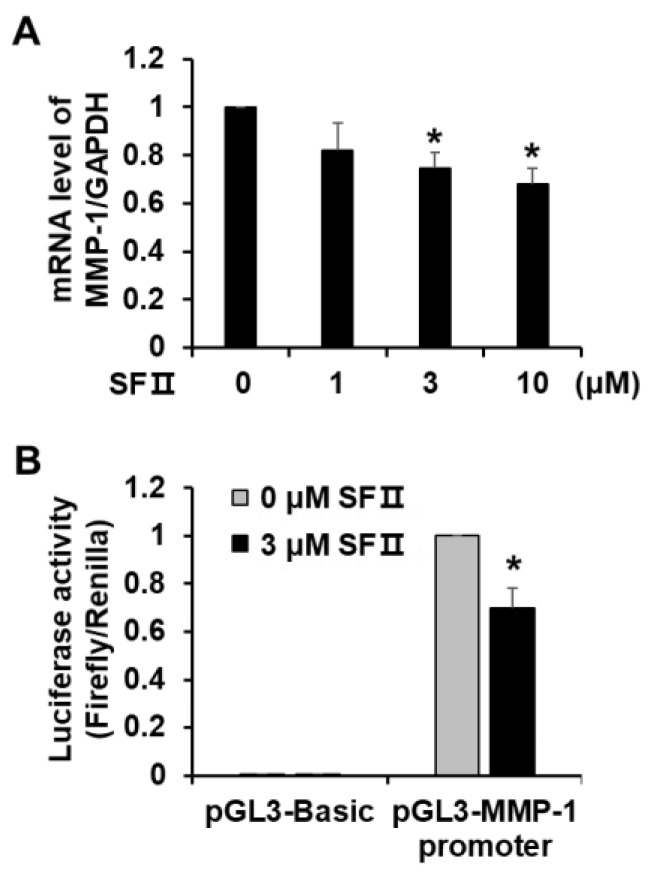
Effect of skullcapflavone II on MMP-1 transcription in foreskin fibroblasts. (**A**) *MMP-1* mRNA levels in DMSO- or skullcapflavone II-treated foreskin fibroblasts were determined using real-time RT-PCR. * *p* < 0.05 vs. the sample incubated with 0 μM skullcapflavone II. (**B**) Foreskin fibroblasts were transiently co-transfected with pGL3-*MMP-1* promoter and pRL-TK as a transfection control. Cells were treated overnight with 3 μM skullcapflavone II. Luciferase activity was determined as the ratio of firefly/*Renilla* luciferase activity. The activity of the *MMP-1* promoter in the presence of skullcapflavone II relative to that in the absence of skullcapflavone II is shown. **p* < 0.05 vs. the sample incubated with 0 μM skullcapflavone II.

**Figure 4 ijms-20-02734-f004:**
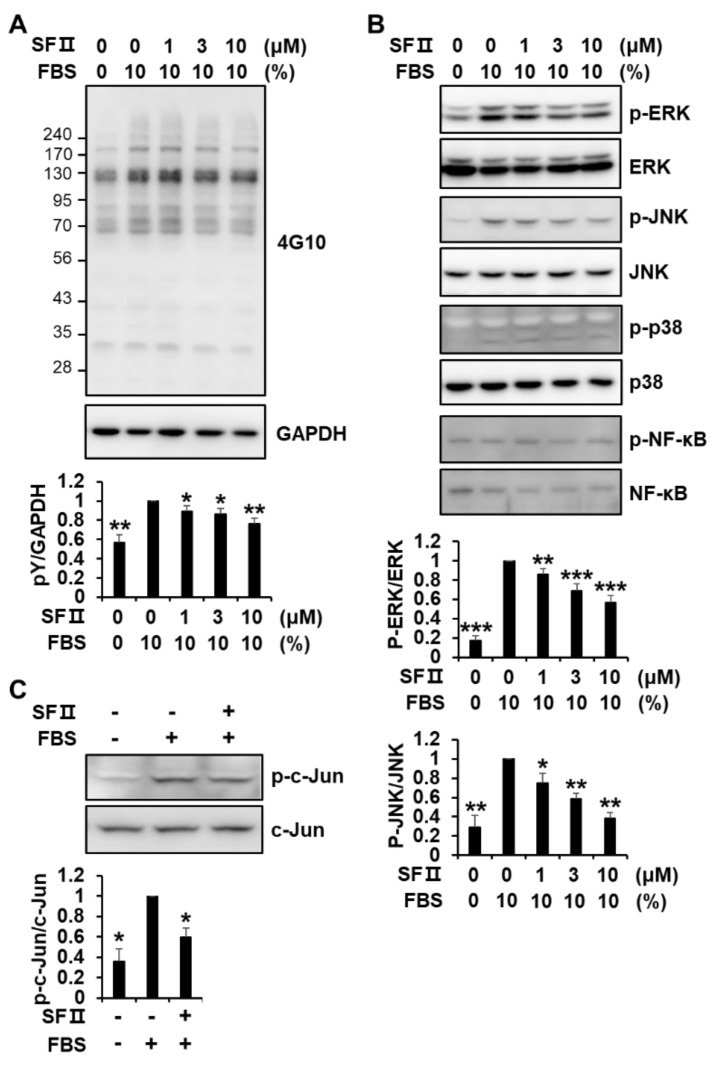
Effect of skullcapflavone II on phosphorylation of signaling molecules in foreskin fibroblasts. Sub-confluent foreskin fibroblasts were starved for 24 h. (**A,B**) Cells were pre-incubated for 30 min with indicated concentrations of skullcapflavone II and then stimulated for 10 min with 10% FBS. Cell lysates were subjected to 9% SDS-PAGE and Western blot analysis with anti-phosphotyrosine (4G10) (A), anti-phospho-extracellular signal-regulated kinase (ERK), anti-phospho-c-Jun N-terminal kinase (JNK), anti-phospho-p38 mitogen-activated protein kinases (MAPK), and anti-phospho-nuclear factor kappa light chain enhancer of activated B cells (NF-κB) p65 antibodies (**B**). (**C**) Cells were pre-incubated for 30 min with (+) or without (−) 3 μM skullcapflavone II, and then FBS was added to a final concentration of 10% and incubated for 30 min. Cell lysates were analyzed by Western blotting with anti-phospho-c-Jun and anti-c-Jun antibodies. * *p* < 0.05, ** *p* < 0.01, and ****p* < 0.001 vs. the sample incubated with 10% FBS alone.

**Figure 5 ijms-20-02734-f005:**
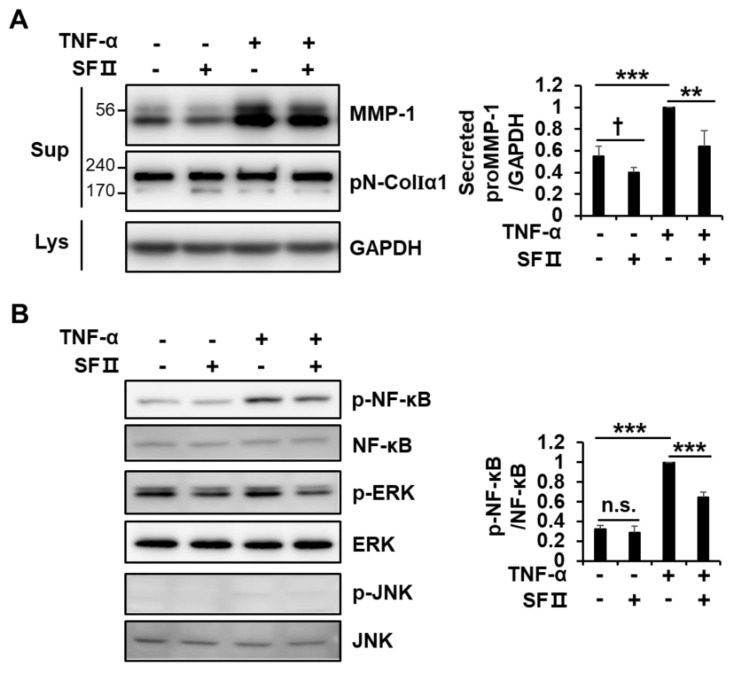
Effect of skullcapflavone II on tumor necrosis factor (TNF)-α-induced MMP-1 expression in foreskin fibroblasts. Foreskin fibroblasts were incubated for 24 h in a serum-free medium with (+) or without (-) 3 μM skullcapflavone II and with (+) or without (−) 1 ng/mL TNF-α. (**A**) The serum-free conditioned medium and cell lysates were analyzed by 9% SDS-PAGE and Western blot with anti-MMP-1, anti-pN-ColIα1, and anti-GAPDH antibodies. (**B**) Cell lysates were analyzed by Western blotting using the indicated antibodies. ** *p* < 0.01 and *** *p* < 0.001 vs. the sample incubated with TNF-α alone; † *p* < 0.05 vs. the sample incubated without TNF-α and without skullcapflavone II; n.s., not significant.

**Figure 6 ijms-20-02734-f006:**
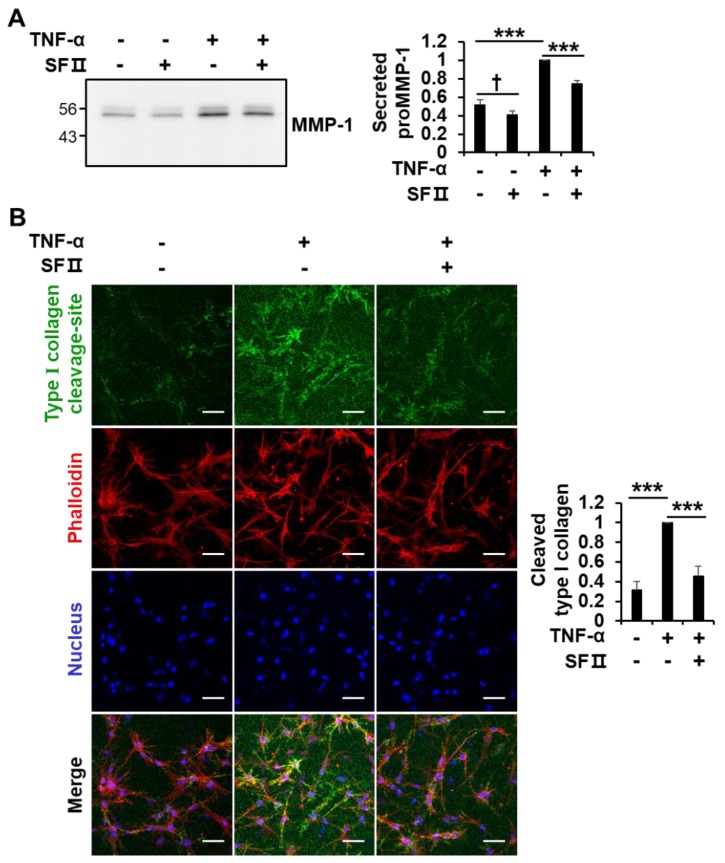
Effect of skullcapflavone II on TNF-α-induced type I collagen degradation in 3D culture of foreskin fibroblasts. Foreskin fibroblasts were embedded within a 3D type I collagen matrix. After polymerization for 1 h at 37 °C, cells embedded in the collagen matrix were incubated for 24 h in serum-free DMEM with (+) or without (-) 3 μM skullcapflavone II and with (+) or without (−) 1 ng/mL TNF-α. (**A**) The conditioned medium in the 3D culture was analyzed by 9% SDS-PAGE and Western blot with anti-MMP-1 antibody. *** *p* < 0.001 vs. the sample incubated with TNF-α alone; † *p* < 0.05 vs. the sample incubated without TNF-α and without skullcapflavone II (**B**) The 3D matrix containing foreskin fibroblasts was stained with anti-type I collagen cleavage-site antibody and Alexa Fluor^®^ 488 goat anti-rabbit IgG (H+L), phalloidin–rhodamine and Hoechst 33258, and cells were analyzed by confocal fluorescence microscopy (200×). Scale bar, 50 μm. The quantification of cleaved type I collagen (Alexa Fluor® 488) relative to nuclear staining (Hoechst 33258) obtained from six randomly chosen fields is shown as the mean ± S.D. *** *p* < 0.001 vs. the sample incubated with TNF-α alone.

**Figure 7 ijms-20-02734-f007:**
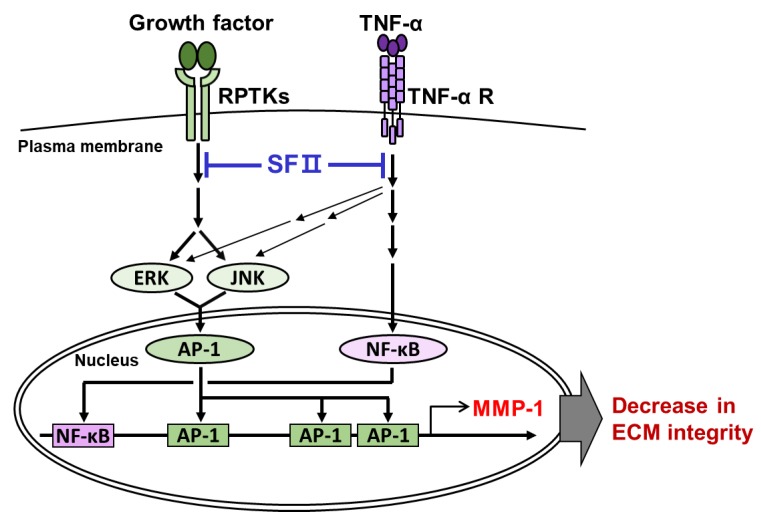
A proposed model describing the role of skullcapflavone II in collagenolysis inhibition. Growth factors in serum activate the transcription factor activator protein-1 (AP-1) through activation of the ERK and JNK pathways. TNF-α activates the transcription factor NF-κB as well as AP-1. Skullcapflavone II inhibits serum- and TNF-α-induced activation of AP-1 and NF-κB, which are required for MMP-1 expression. Skullcapflavone II thus maintains the integrity of extracellular matrix by suppressing MMP-1 expression.

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
