# Peer review of "Skullcapflavone II Inhibits Degradation of Type I Collagen by Suppressing MMP-1 Transcription in Human Skin Fibroblasts"

_ijms, 2019, doi:10.3390/ijms20112734_

Round 1

Reviewer 1 Report

The main problem with this manuscript is that the effect of  Scullcapflavone  on dermal fibroblasts is very weak and some claims about its  activity are not substantiated by the results. The dermal fibroblasts used in this study have not been described, regarding the source, were they primary cells or an immortal cell line, at what passage and confluence were the cells used. Because of the marginal effect, which was shown only in cells of a single source, the experiments have to be reproduced in at least one additional source of dermal fibroblasts. This is essential before publishing and for the conclusions in the manuscript. Below are other specific concerns:

1.      In fig 1A it was shown that Scullcapflavone decreased MMP-1 secretion by dermal fibroblasts, but it did not have an effect on the level of type I collagen. Why was not accumulation of type I collagen increased when MMP-1 had been reduced? In fig 6B, it is shown that SFII decreased cleavage of type I collagen by reducing MMP-1, but fig 1A did not show change in size of type I collagen molecules as the evidence of such cleavage.  Which result is correct; no change in collagen or inhibited degradation?

2.      In fig 1B it is claimed that SFII reduced MMP-1 expression at 48h in cells grown in 3% serum. I don’t see any reduction. Likewise, for fig 4 A it is claimed that serum induced tyrosine phosphorylation is significantly reduced by SFII. I don’t see any reduction and the experiment was not quantified.

3.      Fig 4c shows reduction of p-c-Jun after SFII treatment which was only ~15%. How can the western blots analysis reliably measure changes in protein expression of 15%?  This is also the problem with fig 4B; the changes caused by SFII are so marginal that they are difficult to quantify using western blots. Therefore, it is essential to repeat these experiments in a different isolate of dermal fibroblasts to confirm the findings. Without additional validation the claim that SFII “could be useful in therapies for maintaining the integrity of extracellular matrix” can not be made.

Author Response

Responses to the Reviewer 1’s Comments

General Comment: The main problem with this manuscript is that the effect of Scullcapflavone  on dermal fibroblasts is very weak and some claims about its  activity are not substantiated by the results. The dermal fibroblasts used in this study have not been described, regarding the source, were they primary cells or an immortal cell line, at what passage and confluence were the cells used. Because of the marginal effect, which was shown only in cells of a single source, the experiments have to be reproduced in at least one additional source of dermal fibroblasts. This is essential before publishing and for the conclusions in the manuscript.

Response to the General Comment:

According to the Reviewer 1’s comment, we described the source, passage range, and confluence of primary human foreskin fibroblasts in the Materials and Method section (lines 337-344).

In addition, we analysed the effect of skullcapflavone II on the expression and MMP-1 and type I collagen using human dermal fibroblasts from the buttock skin of young individual (lines 338-339). We found that skullcapflavone II had the same effect on the secretion of MMP-1 in buttock dermal fibroblasts as in foreskin fibroblasts. We have included this result in the right panel of the Figure 1A and described in the Results sections (lines 72-79) in the revised manuscript.

Moreover, we performed the statistical analysis for the results of Figures 1B, 4A, 4B, and 6B, which were obtained from three or more independent experiments. We have shown that these results are statistically significant. We therefore believe that our conclusions are more convincing than they were before the revision.

Comment 1: In fig 1A it was shown that Scullcapflavone decreased MMP-1 secretion by dermal fibroblasts, but it did not have an effect on the level of type I collagen. Why was not accumulation of type I collagen increased when MMP-1 had been reduced? In fig 6B, it is shown that SFII decreased cleavage of type I collagen by reducing MMP-1, but fig 1A did not show change in size of type I collagen molecules as the evidence of such cleavage.  Which result is correct; no change in collagen or inhibited degradation?

Response 1:

It was known that human fibroblasts secrete MMP-1 as a pro-form; mostly unglycosylated (52 kDa) and partly glycosylated (57 kDa) [ref. 20 in the revised manuscript]. In our experiment, the secreted MMP-1 was detected at 52 kDa (Figure 1A), suggesting that it should be pro-MMP-1. ProMMP-1 in the 2D culture medium cannot be activated into the mature active form (42 kDa), due to the dilution of the proMMP-1 into a large volume of medium. Thus, the secreted proMMP-1 cannot cleave type I collagen in the medium. We showed that skullcapflavone II reduced the secretion of MMP-1 in Figure 1A. Nevertheless, the level of type I collagen in the medium was not increased since no mature MMP-1 was present in the medium.  

To analyse whether skullcapflavone II affects the degradation of type I collagen by down-regulating MMP-1 expression, we used a 3D collagen matrix system for culturing foreskin fibroblasts in, mimicking connective tissue. As shown in Figure 6B, we have measured the cleavage of type I collagen by active form of MMP-1 using a specific antibody that detects a cleaved fragment of type I collagen. We found that skullcapflavone II decreased MMP-1 expression and degradation of type I collagen by mature MMP-1 processed in the matrix.

Therefore, our results for Figure 1A and Figure 6B are all correct. In the 2D culture medium, the decreased secretion of MMP-1 cannot reduce the level of type I collagen. However, in the 3D culture matrix, decreased expression of MMP-1 is able to decrease the degradation of type I collagen.

We have explained the underlying information and our findings and interpretations in the Results (lines 72-79 and 198-202) and Discussion (lines 279-280 and 284-288) sections in the revised manuscript.

Comment 2: In fig 1B it is claimed that SFII reduced MMP-1 expression at 48h in cells grown in 3% serum. I don’t see any reduction. Likewise, for fig 4 A it is claimed that serum induced tyrosine phosphorylation is significantly reduced by SFII. I don’t see any reduction and the experiment was not quantified.

Response 2:

To demonstrate the reproducibility of our results, we repeated the experiments for Figures 1B and 4A and performed a statistical analysis of the quantified data from three or more independent experiments. As a result, we have shown that skullcapflavone II significantly decreased the secretion of MMP-1 in cells incubated with no serum and 3% serum, in Figure 1B. In addition, skullcapflavone II significantly reduced serum-induced tyrosine phosphorylation of cellular proteins and activation of ERK and JNK, in Figures 4A and 4B. Therefore, we now convincingly show that skullcapflavaone II down-regulates MMP-1 expression by blocking the activation of signalling pathways such as ERK and JNK. The graphs for statistical analysis in Figures 1B, 4A, and 4B were included in the revised manuscript.

Comment 3: Fig 4c shows reduction of p-c-Jun after SFII treatment which was only ~15%. How can the western blots analysis reliably measure changes in protein expression of 15%?  This is also the problem with fig 4B; the changes caused by SFII are so marginal that they are difficult to quantify using western blots. Therefore, it is essential to repeat these experiments in a different isolate of dermal fibroblasts to confirm the findings. Without additional validation the claim that SFII “could be useful in therapies for maintaining the integrity of extracellular matrix” cannot be made.

Response 3:

We quantified the band intensities of the western blots with the widely used MultiGauge software, as added in the Materials and Methods section in the revised manuscript (lines 358-359). To increase the accuracy of the quantification, the band intensity was calibrated by taking the background signal into account and dividing the GAPDH level for normalization. In addition, we performed a statistical analysis of the data from at least three independent experiments. As a result, we have included the statistical analysis of western blot data for Figures 1B, 4A and 4B, and updated it for Figure 4C, in the revised manuscript.

As described in the Response to the General Comment, we analysed the effect of skullcapflavone II on the expression and MMP-1 and type I collagen using human dermal fibroblasts from the buttock skin of young individual (lines 337-339) in addition to human foreskin fibroblasts. We found that skullcapflavone II had the same effect on the secretion of MMP-1 in buttock dermal fibroblasts as it did in foreskin fibroblasts. We have included this result in the right panel of the Figure 1A and described in the Results sections (lines 72-79) in the revised manuscript.

Through this validation, we now convincingly propose that skullcapflavone II should be useful in therapies to maintain the integrity of extracellular matrix.

Reviewer 2 Report

The authors give a  novel study of skullcapflavone II inhibiting degradation of Type I collagen by suppressing MMP-1 Transcription in Human Skin Fibroblasts  with clearly collected and explained results. The main criticism perhaps is the lack of explanation why Collagen Type I was selected over other collagen types and tissues of interest (where other collagens are present). Text in lines 44- 57 requires more referenced work on collagen overall. I am wondering why 10uM of SFII was the highest used concentration in  the exp in Fig 1?   What are the units on the Y-axis in Fig 2A? Could  authors indicate p* values in the Figures, as well? Could the authors elaborate more on the statistics and data analysis, overall ( line 360-363)?  It would be recommended to list mRNA level per number of cells in Fig 3A. Confocal images in Fig 6 need to contain the scale bar and that detail should be indicated in the Figure's legend as an info, as well. Was the scale bar 10um? What was the statistics, how many images were acquired for the conclusions made related to the imaging Figure 6? The line 247- It would be recommended to back up " Breakdown of Collagen by MMP-1..." statement with more referenced work. Line 302- For how many days the cells were cultured under the described conditions? There is a missing section on the Confocal imaging part within Material and Methods. Was the averaging of images performed in order to enhance signal to noise ratio? Was the check for the oversaturated pixels performed during the image acquisition procedure? That needs to be added as well. Fig 6 may contain some of the control experiments shown, as well. How was the spillover prevented during the image acquisition in Fig 6? The font in line 190 is larger than in the rest of the text in that Figure legend.

Author Response

Responses to the Reviewer 2’s Comments

General Comment: The authors give a novel study of skullcapflavone II inhibiting degradation of Type I collagen by suppressing MMP-1 Transcription in Human Skin Fibroblasts  with clearly collected and explained results.

Comment 1: The main criticism perhaps is the lack of explanation why Collagen Type I was selected over other collagen types and tissues of interest (where other collagens are present). Text in lines 44- 57 requires more referenced work on collagen overall.

Response 1:

As suggested by Reviewer 2, we briefly described subfamilies of collagen and collagen types cleaved by MMP-1 with references, in the Introduction section of the revised manuscript (lines 44-49 and 55-58). In addition, we briefly stated in the revised manuscript (lines 61-63) that type I collagen was selected in our study because it is most abundant among collagen types, highly expressed in fibroblasts, and cleaved by MMP-1.

Comment 2: I am wondering why 10 uM of SFII was the highest used concentration in the exp in Fig 1?

Response 2:

At the beginning of our experiments, we analysed the effect of skullcapflavone II on the growth of fibroblasts at concentrations up to 100 μM. Cell growth was greatly reduced in the presence of 100 μM skullcapflavone II (up to 47% decrease in growth at 2 days). Our experiment focused more on the effect of ECM modulation by skullcapflavone II than on the effect of cell growth. Therefore, we selected 10 μM of skullcapflavone II, which showed a slight decrease (up to 15%) in cell growth, as the highest concentration of skullcapflavone II in our experiments.

Comment 3: What are the units on the Y-axis in Fig 2A?

Response 3:

The number of viable cells was measured by MTT assay as described in the Materials and Methods section (lines 363-365). Cell growth in the Y axis of Figure 2A is shown as a fold change in the number of viable cells in the presence of skullcapflavone II relative to that in the absence of skullcapflavone II. This information has been described in the legend of Figure 2 (lines 107-109) in the revised manuscript.

Comment 4: Could authors indicate p* values in the Figures, as well?

Response 4:

We have added the definition of p values in the legends of all the Figures in the revised manuscript.

Comment 5: Could the authors elaborate more on the statistics and data analysis, overall (line 360-363)?

Response 5:

We have described data acquisition from band intensities of western blots and fluorescence intensities of microscopic images (lines 358-359 and 401-403), and statistical analysis (lines 405-408) in the Materials and Methods sections of the revised manuscript.

Comment 6: It would be recommended to list mRNA level per number of cells in Fig 3A.

Response 6:

As suggested by Reviewer 2, we have estimated the number of MMP-1 mRNA molecules per cells using MMP-1 cDNA as a reference. The MMP-1 mRNA molecules per cell were measured to be 40, 32.5, 29.6, and 27.2 at 0, 1, 3, and 10 μM skullcapflavone II, respectively. We included this result in the Results section of the revised manuscript (lines 116-118), as follows:

From the reference curve generated by serial dilution of MMP-1 cDNA, the MMP-1 mRNA molecules per cell were estimated to be 40.0 at 0 uM skullcapflavone II and 27.2 at 10 uM skullcapflavone II.

Comment 7: Confocal images in Fig 6 need to contain the scale bar and that detail should be indicated in the Figure's legend as an info, as well. Was the scale bar 10um?

Response 7:

We have inserted a scale bar (50 μm) into Figure 6B and described it in the legend of Figure 6 (line 220) in the revised manuscript.

Comment 8: What was the statistics, how many images were acquired for the conclusions made related to the imaging Figure 6?

Response 8:

To avoid bias during image acquisition, six images per each sample were obtained from randomly selected fields. Quantification of fluorescence intensity was determined using Image J software. A statistical analysis of the fluorescence intensity of cleaved type I collagen relative to nuclear staining was added as a graph in Figure 6B. We also described the methods for image acquisition and statistical analysis in the Materials and Methods section (lines 401-403) and the legend of Figure 6B (lines 220-222) of the revised manuscript.

Comment 9: The line 247- It would be recommended to back up " Breakdown of Collagen by MMP-1..." statement with more referenced work.

Response 9:

We have added references for this information in the revised manuscript (line 279).

Comment 10: Line 302- For how many days the cells were cultured under the described conditions?

Response 10:

We cultured fibroblasts subconfluently and newly plated cells were incubated overnight for attachments prior to use. The sources, culture conditions, and passage numbers of human foreskin fibroblasts and human buttock dermal fibroblasts were described in the Materials and Methods section of the revised manuscript (lines 337-344).

Comment 11: There is a missing section on the Confocal imaging part within Material and Methods. Was the averaging of images performed in order to enhance signal to noise ratio? Was the check for the oversaturated pixels performed during the image acquisition procedure? That needs to be added as well.

Response 11:

We used 16 times of frame averaging and setting below the saturation levels during the image acquisition. We have briefly added the method for the confocal fluorescence microscopy and image acquisition in the Materials and Methods section of the revised manuscript (lines 398-403).

Comment 12: Fig 6 may contain some of the control experiments shown, as well.

Response 12:

We agree with Review 2’s comment that some of the control experiments, such as knockdown of MMP-1 in fibroblasts, would help us to demonstrate our conclusion. Because fibroblasts had low transfection efficiency and it took some time to acquire an efficient MMP-1 knockdown system, it was not possible to conduct such a control experiment within a limited revision period.

Instead, we performed the statistical analysis of the relative fluorescence intensities of the randomly selected images in Figure 6B of the revised manuscript.

With the improved results from this revision, we can more convincingly suggest that skullcapflavone II should be useful in therapies to maintain the integrity of extracellular matrix.

Comment 13: How was the spillover prevented during the image acquisition in Fig 6?

Response 13:

We used excitation wavelengths of 405 nm for Hoechst 33258, 488 nm for Alexa Fluor® 488, and 555 nm for rhodamine. The spectrum ranges were 360-460 nm for Hoechst 33258, 490-520 nm for Alexa 488, and 560-590 nm for rhodamine. We therefore assume that there is no spillover during image acquisition for Figure 6.

Comment 14: The font in line 190 is larger than in the rest of the text in that Figure legend.

Response 14:

We fixed the font size in the legend of Figure 6 of the revised manuscript.

Round 2

Reviewer 1 Report

The inclusion of an additional isolate of skin fibroblasts improved the manuscript. New quantification of western blots has also been included. However, the quantification of changes in protein expression of 20-30% by western blots is dubious (check critical evaluation of western blots in BioMed Research International, Volume 2019, Article ID 5214821, https://doi.org/10.1155/2019/5214821). Although the new statistical analysis of the western blots showed a significant change, the overall effect of SFII is weak and the claim that “we now convincingly propose that skullcapflavone II should be useful in therapies to maintain the integrity of extracellular matrix” is an exaggeration. However, if the editor agrees to publish this manuscript, I will have no further objections.

Reviewer 2 Report

All the comments were addressed. Thanks.